# Stochastic dynamics of *Francisella tularensis* infection and replication

**Jonathan Carruthers**[1], **Grant Lythe**[1], **Martín López-García**[1], **Joseph Gillard**[2], **Thomas R. Laws**[2], **Roman Lukaszewski**[2], **Carmen Molina-París**[1]*

**1** Department of Applied Mathematics, University of Leeds, Leeds, United Kingdom, **2** CBR Division, Defence Science and Technology Laboratory, Salisbury, United Kingdom

* carmen@maths.leeds.ac.uk

**Data Availability Statement:** Computer codes (in Python) to generate the numerical realisations of the agent-based model and to perform the cohort

## Abstract

We study the pathogenesis of *Francisella tularensis* infection with an experimental mouse model, agent-based computation and mathematical analysis. Following inhalational exposure to *Francisella tularensis* SCHU S4, a small initial number of bacteria enter lung host cells and proliferate inside them, eventually destroying the host cell and releasing numerous copies that infect other cells. Our analysis of disease progression is based on a stochastic model of a population of infectious agents inside one host cell, extending the birth-and-death process by the occurrence of catastrophes: cell rupture events that affect all bacteria in a cell simultaneously. Closed expressions are obtained for the survival function of an infected cell, the number of bacteria released as a function of time after infection, and the total bacterial load. We compare our mathematical analysis with the results of agent-based computation and, making use of approximate Bayesian statistical inference, with experimental measurements carried out after murine aerosol infection with the virulent SCHU S4 strain of the bacterium *Francisella tularensis*, that infects alveolar macrophages. The posterior distribution of the rate of replication of intracellular bacteria is consistent with the estimate that the time between rounds of bacterial division is less than 6 hours *in vivo*.

## Author summary

Infecting a host cell is required for the replication of many types of bacteria and viruses. In some cases, infected cells release new infectious agents continuously over their lifetime. In others, such as the *Francisella tularensis* bacterium studied here, they are released in a single burst that coincides with the cell's death. We show how a stochastic model, the birth-and-death process with catastrophe, can be used to characterise infection in a single cell, thereby allowing us to account for burst events and quantify the kinetics of pathogenesis in the lung, the initial site of infection, as well as in other organs that the infection spreads to. We learn about the parameters of the mathematical model of *Francisella tularensis* infection making use of the experimental measurements of bacterial loads, together with approximate Bayesian statistical inference methods. The most important parameter describing the pathogenesis is the rate of replication of intracellular bacteria.

analysis are available in this link http://archive.researchdata.leeds.ac.uk/677/.

**Funding:** The experimental work shown here was funded by the UK Ministry of Defence. Mathematical research was supported by the Medical Research Council through the Skills Development Fellowship number MR/N014855/1 MLG. The funders had no role in study design, data collection and analysis, decision to publish, or preparation of the manuscript.

**Competing interests:** The authors have declared that no competing interests exist.

## Introduction

*Francisella tularensis*, the causative agent of tularemia, is extremely infectious and considered a biothreat agent [1–3]. Treatment options are limited: a live attenuated vaccine exists but is not in mainstream use [4, 5]. Protection via antibiotics is dependent on early diagnosis and timely administration [6]. *Francisella tularensis* bacteria may be inhaled in an aerosol, with initial doses as low as ten colony-forming units (CFU) resulting in respiratory or pneumonic tularemia [7–10]. The bacteria enter alveolar macrophages [11–16], evading initial immune recognition and inflammatory response because of their atypical lipopolysaccharide [17]. They are able to escape from phagosomes in less than an hour and, as illustrated in Fig 1, begin multiple rounds of replication in the cytosol [18–21]. Instead of producing inflammatory cytokines, the first infected macrophages produce anti-inflammatory TGF-$\beta$ cytokine. The eventual death of the host macrophage [22] returns bacteria to the extracellular environment, from where they can migrate to another organ, or again infect macrophages in the lung.

Our agent-based computational model of the first 72 hours after infection is based on that of Gillard *et al.* [23], who considered the first 24 hours after infection. Classical mathematical models of intracellular infection consider variables describing populations of uninfected cells, infected cells and free infectious particles [24–26]. In such models, the rate of production of new infected cells is assumed to be proportional to the number of infected cells, which is true if each infected cell, independently, releases infectious particles at a constant rate. It is possible to go beyond the simplest hypothesis by considering subpopulations of infected cells: in an "eclipse" phase or productive phase [27, 28], or considering different multiplicities of infection and co-infection [29]. In this work, we seek to describe an scenario where bacteria continue to divide inside the host cell, without any being released from the cell, until the host cell ultimately ruptures, typically releasing more than a hundred bacteria at a time [23, 30–33]. Our

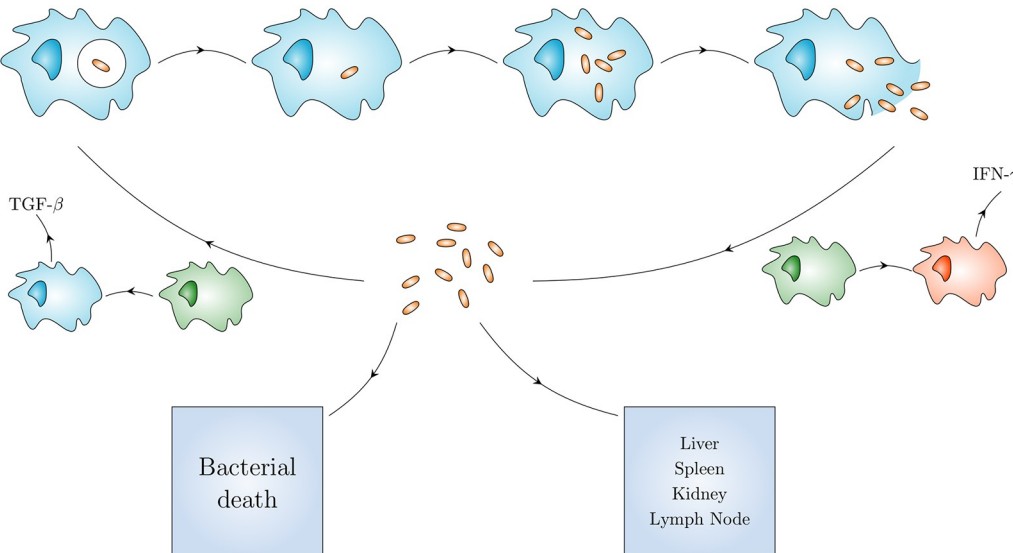

**Fig 1. Within-host model of *Francisella tularensis* pathogenesis.** A single bacterium is taken up by a macrophage, inside a phagosome (top left). Without activating the macrophage, the bacterium escapes and multiplies inside the cytosol (top line) eventually causing the macrophage to rupture and release many bacteria. Free bacteria (center) may infect other macrophages, die or migrate to a different organ (bottom line). Macrophages exist in resting (green), anti-inflammatory (blue) and pro-inflammatory (red) states.

mathematical approach is a stochastic model of the population of infectious agents inside one host cell, extending the simple birth-and-death process by the occurrence of catastrophes.

A consequence of the extreme virulence of *Francisella tularensis* is that initial doses of bacteria used in experiments are small enough for it to be reasonable to assume that host macrophages are infected by only one bacterium each [18]. Thus, the ensemble of realisations of the stochastic process describing the dynamics of a population of bacteria inside one cell can be thought of as describing the dynamics inside a set of host cells, that behave independently until they rupture.

In this paper we model the number of bacteria inside an infected macrophage, making use of a birth-and-death process with catastrophe [34], and the bacterial populations in multiple organs in the first few days after infection. Expressions for these variables are first computed for the lung before considering the mesenteric lymph nodes (MLNs), liver, kidney and spleen. We describe host cell rupture by a load-dependent "hazard rate": an infected macrophage's rupture probability per unit time is proportional to its bacterial load. Thus, we assume that cells with high bacterial load at a given time are more likely to rupture than those with a lower bacterial load, but there is no fixed maximum or minimum load [35]. Our assumption is consistent with observations of infection, and apoptosis, of murine macrophage-like J774.A1 cells [36]. Alternatively, a mathematical modeller may take the rupture time and the number of bacteria released per rupture event to be fixed parameters that can be inferred from experimental data [30], or assume a distribution of rupture times that is independent of the intracellular dynamics [33]. Similar questions arise in the modelling of *Salmonella enterica* infections [37–39]. In this work, we calculate the distributions of rupture times and number of bacteria released per rupture event as a consequence of the stochastic description of an infected macrophage and its bacterial contents.

We make use of the Sobol method of global sensitivity analysis to identify which parameters, in the mathematical model that describes the early days of infection, have the greatest effect on bacterial counts in each organ [40, 41]. Using a decomposition of the variance, this approach allows us to see how the variance in bacterial counts changes when combinations of parameters are fixed. If fixing a parameter results in a large reduction in the variance, this parameter will be of greater importance. With an approximate Bayesian computation algorithm [42], we learn about the most important parameters by comparing predictions of the model with the bacterial counts measured in the experiments.

## Results

### Birth-death-catastrophe process

In order to describe the course of *Francisella tularensis* infection for a given host, we need to consider the dynamics inside one infected cell, beginning at the time of entry of one infectious agent into the cytosol and ending either with the rupture of the cell and release of its bacterial content, or with the elimination of the infection from the cell. To describe the dynamics inside a host cell, we assume the most simple hypothesis; namely, that each bacterium has a constant division and death rate. The corresponding rates for the bacterial population inside a single host cell are proportional to its size. Since the probability per unit time that the host cell ruptures is a function of its intracellular bacterial population size, the independence property required for a branching process description does not hold in this case. Rupture of the host cell is a "catastrophe" event that affects every bacterium in the cell at the same time. Catastrophes have been considered mathematically as an extension of birth-and-death processes [43, 44], including scenarios where a subset of the population is removed [45–51], or where a catastrophic event kills the entire population [52–57]. Our interest here is in the process depicted

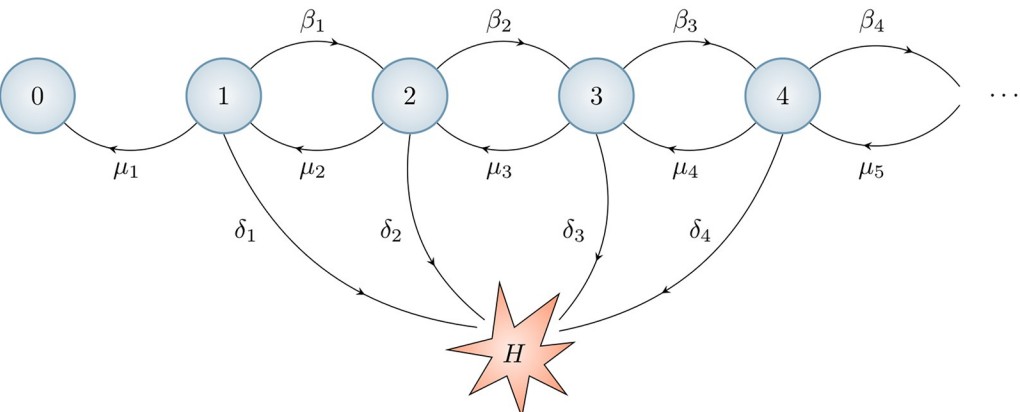

**Fig 2. A birth-and-death process with catastrophe representing division, death and rupture.** The state $n$ represents a macrophage with $n$ cytosolic bacteria. There are three types of events: transition to state $n + 1$ (division of a bacterium, rate $\beta_n$), transition to state $n$-1 (death of a bacterium, rate $\mu_n$), and transition to state $H$ (rupture of the macrophage with release of $n$ bacteria, rate $\delta_n$). In this work we assume that $\beta_n = \beta n$, $\mu_n = \mu n$ and $\delta_n = \delta n$. The states 0 and $H$ are absorbing. The infected macrophage survives for as long as it does not reach the state $H$.

in Fig 2, where two distinct absorbing states exist, both of which represent the loss of all intracellular bacteria. The first, state 0, represents the recovery of the macrophage due to successful elimination of its bacterial load. The second, state $H$, represents the rupture of the macrophage and the release of its entire content of bacteria [34]. Thus, the number of bacteria inside the cytosol of a single infected cell is denoted $\mathbf{X}_t$ where

$$\mathbf{X}_t \in \{0, 1, 2, \ldots\} \cup \{H\} \ .$$

In this Section, we take $\mathbf{X}_0 = 1$. That is, we assume that the cell only phagocytoses one bacterium, and $t = 0$ is the time at which the bacterium escapes from the phagosome. We also ignore, for now, the possibility of reinfection. Macrophages containing any number of bacteria may rupture; those with high bacterial load are more likely to do so than those with low bacterial loads [58]. Thus, we assume that the rate associated with the catastrophe event is proportional to the instantaneous number of bacteria: if a macrophage contains $\mathbf{X}_t$ bacteria at time $t$, the probability that it ruptures before time $t + \Delta t$ is $\delta \mathbf{X}_t \Delta t$, in which case all of the $\mathbf{X}_t$ bacteria are released from the cell.

We assume that *Francisella tularensis* bacteria replicate in the cytosol of their host macrophages with rate $\beta$ per bacterium, and are susceptible to intracellular death through misfortune or cellular defence mechanisms with rate $\mu$. Estimates based on observations of *Francisella tularensis* proliferation suggest that $\beta \simeq 0.16 \ h^{-1}$ and $\mu \ll \beta$ [23, 59]. As with cell rupture, the probability that an intracellular bacterium divides or dies in a short interval $(t, t + \Delta t)$ is proportional to $\mathbf{X}_t$. Thus, if $\mathbf{X}_t$ is the bacterial load at time $t$, then the bacterial load at time $t + \Delta t$ is either

$$
\begin{array}{ll}
\mathbf{X}_t & \text{with probability } 1 - (\mu + \beta + \delta)\mathbf{X}_t \Delta t, \\
\mathbf{X}_t + 1 & \text{with probability } \beta \mathbf{X}_t \Delta t, \\
\mathbf{X}_t - 1 & \text{with probability } \mu \mathbf{X}_t \Delta t, \\
H \ (\text{rupture}) & \text{with probability } \delta \mathbf{X}_t \Delta t.
\end{array}
$$

## Dynamics of bacterial load in a single infected cell

The first quantity of interest from the perspective of an infected macrophage is its survival function, $S(t)$: the probability that a single infected macrophage has not ruptured by time $t$. $S(t)$ is also the fraction of macrophages with a single bacterium in their cytosol at time $t = 0$ that survive to time $t$. We can think of this as the surviving fraction in a large group of macrophages, each initially infected with a single bacterium. We can write

$$S(t) = \mathcal{P}[\text{macrophage survives to time } t | \mathbf{X}_0 = 1] = \mathcal{P}[\mathbf{X}_t \neq H | \mathbf{X}_0 = 1] .$$

If a macrophage is carrying $\mathbf{X}_t$ bacteria in its cytosol at time $t$, its probability of rupture between $t$ and $t + \Delta t$ is equal to $\delta \mathbf{X}_t \Delta t$. The function $S(t)$ is the average over realisations, so $S(t + \Delta t) - S(t) = \delta \mathbb{E}(\mathbf{X}_t) - \Delta t$. In other words, $S(t)$ satisfies the following differential equation

$$\frac{\mathrm{d}}{\mathrm{d}t} S = -\delta \mathbb{E}(\mathbf{X}_t | \mathbf{X}_0 = 1) . \tag{1}$$

One way to calculate the survival function is to find the distribution of $\mathbf{X}_t$ and make use of (1). Before proceeding to the explicit calculation of $\mathbf{X}_t$, however, it is useful to consider a direct method for calculating $S(t)$, using an extended definition of the survival function. Let $S^{(k)}(t)$ be the survival function of a single macrophage with $k$ cytosolic bacteria at $t = 0$:

$$S^{(k)}(t) = \mathcal{P}[\text{macrophage survives to time } t | \mathbf{X}_0 = k] .$$

If $\mathbf{X}_0 = 1$ then, as $\Delta t \to 0$, either

- $\mathbf{X}_{\Delta t} = 1$ with probability $1 - (\beta + \mu + \delta)\Delta t$,

- $\mathbf{X}_{\Delta t} = 0$ with probability $\mu \Delta t$,

- $\mathbf{X}_{\Delta t} = 2$ with probability $\beta \Delta t$,

- $\mathbf{X}_{\Delta t} = H$ with probability $\delta \Delta t$.

  Thus, we have

$$S(t + \Delta t) = \beta \Delta t \, S^{(2)}(t) + [1 - (\beta + \mu + \delta)\Delta t]S(t) + \mu \Delta t ,$$

and

$$\frac{\mathrm{d}}{\mathrm{d}t} S = \mu - (\beta + \mu + \delta)S + \beta S^{(2)} . \tag{2}$$

If $k = 2$, then the number of bacteria at time $t$ can be written as the sum of two families: the first initial bacterium with its progeny and the second initial bacterium with its progeny (see Fig 3). The probability of rupture between $t$ and $t + \Delta t$ is given by $2S^{(1)}(t)\frac{\mathrm{d}}{\mathrm{d}t}S^{(1)}(t)\Delta t$, with $S^{(1)}(t) = S(t)$. That is

$$\frac{\mathrm{d}}{\mathrm{d}t} S^{(2)}(t) = 2S^{(1)}(t) \, \frac{\mathrm{d}}{\mathrm{d}t} S^{(1)}(t) .$$

We therefore conclude that $S^{(2)}(t) = [S(t)]^2$. Similarly, the probability that a macrophage, initially infected by $k$ bacteria, is alive at time $t$ is equal to the probability that $k$ independent macrophages, each infected by a single bacterium, are all alive at time $t$:

$$S^{(k)}(t) = [S(t)]^k . \tag{3}$$

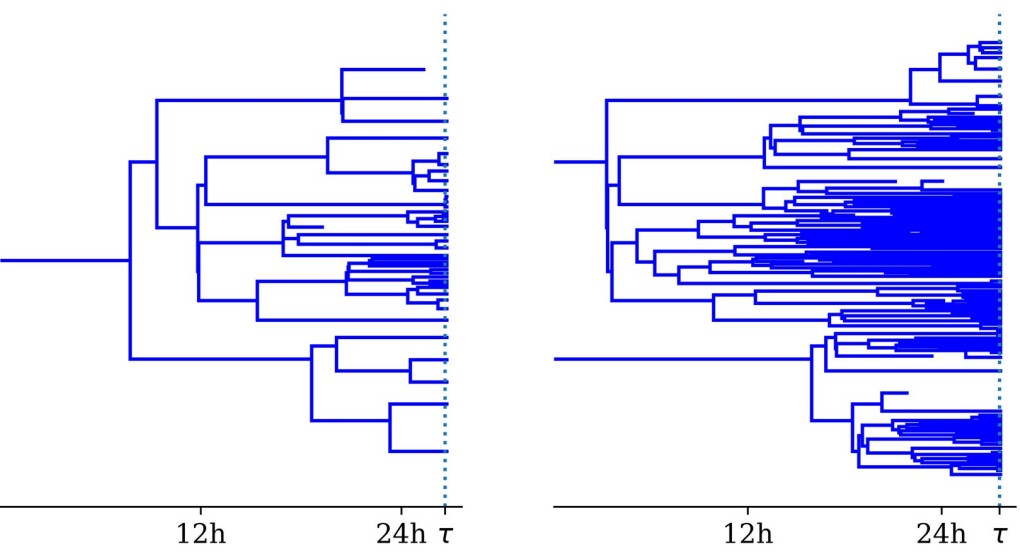

**Fig 3. Realisations of the birth-death process with catastrophe.** On the left, the initial number of bacteria, $k = 1$. On the right, the initial number of bacteria, $k = 2$. On the right, the two families follow, independently, the same stochastic process as in the case $k = 1$. However, the catastrophe affects both families at the same instant. The parameter values are $\beta = 0.15\ h^{-1}$, $\mu = 0.01\ h^{-1}$ and $\delta = 0.01\ h^{-1}$ [23].

Thus, from (2), the survival function $S(t)$ obeys

$$\frac{\mathrm{d}}{\mathrm{d}t}S = \mu - (\beta + \mu + \delta)S + \beta S^2 \ . \tag{4}$$

The solution of (4) agrees with that obtained by Karlin and Tavare [34]

$$S(t) = \frac{a(b-1) + b(1-a)\mathrm{e}^{-\beta(b-a)t}}{b-1+(1-a)\mathrm{e}^{-\beta(b-a)t}} \ , \tag{5}$$

where $a$ and $b$ are the zeros of the function $F(S) = \beta S^2 - (\beta + \mu + \delta)S + \mu$; that is, $ab = \frac{\mu}{\beta}$ and $a + b = \frac{\beta+\delta+\mu}{\beta}$ (see Fig 4). The survival function itself is shown in Fig 5. Note that $S(t) \to a$ as $t \to +\infty$, so that $a$ is equal to the probability that the infected macrophage eliminates the infection, as opposed to rupturing. If $\mu = 0$ then $a = 0$, $b = 1 + \frac{\delta}{\beta}$ and the survival function takes the simpler form,

$$S(t) = \frac{\beta + \delta}{\beta + \delta \mathrm{e}^{(\beta+\delta)t}} \ .$$

The second quantity of interest is $f(t)$, the probability density function of the time until rupture of a macrophage initially infected with a single bacterium, given by

$$f(t) = -\frac{\mathrm{d}}{\mathrm{d}t}S(t) = \frac{\delta(b-a)^2 \mathrm{e}^{-\beta(b-a)t}}{\left(b-1+(1-a)\mathrm{e}^{-\beta(b-a)t}\right)^2} \ .$$

The maximum value of $f(t)$ is found at (see Fig 5)

$$t_{\mathrm{max}} = \frac{\log(\beta/\delta)}{\beta(b-a)} \ . \tag{6}$$

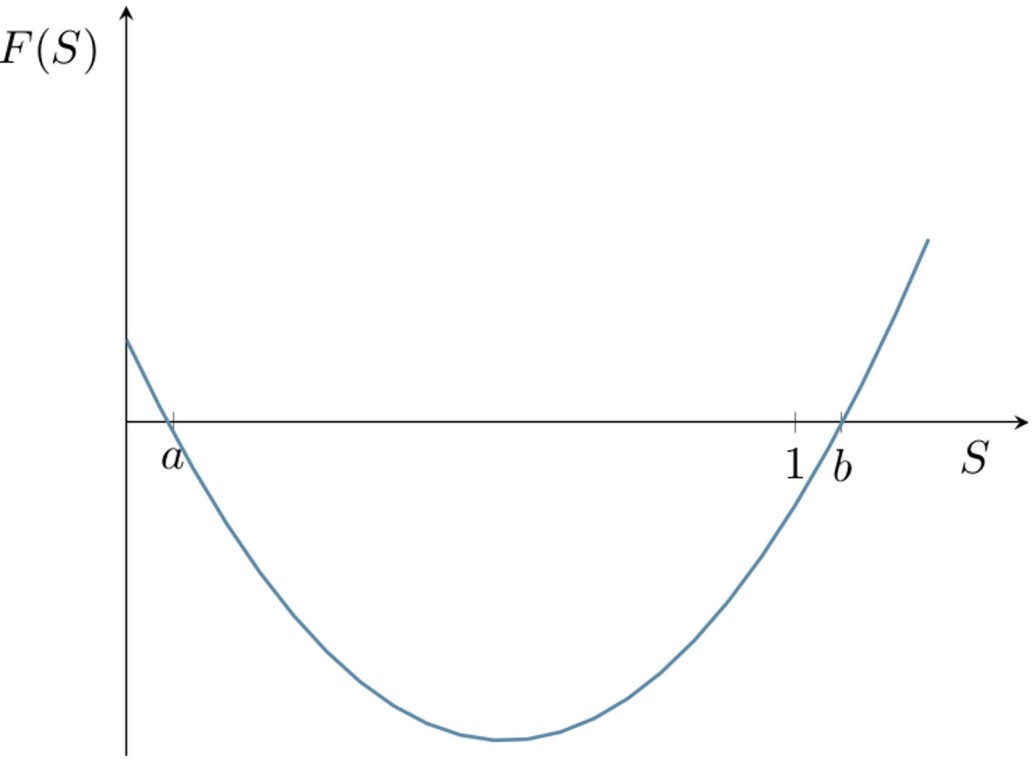

**Fig 4. Polynomial governing the survival function.** $F(S) = \beta S^2 - (\beta + \mu + \delta)S + \mu$ is the RHS of (4) with parameter values $\beta = 0.15\ h^{-1}$, $\mu = 0.01\ h^{-1}$ and $\delta = 0.01\ h^{-1}$. The constants $a$ and $b$ satisfy $ab = \frac{\mu}{\beta}$ and $a + b = \frac{\beta + \delta + \mu}{\beta}$. The value of $a$ is equal to the probability that the infected macrophage eliminates the infection, rather than ruptures.

If $\mu = 0$ then

$$f(t) = \delta \frac{(\beta + \delta)^2 e^{(\beta + \delta)t}}{\left(\beta + \delta e^{(\beta + \delta)t}\right)^2} \ . \tag{7}$$

The third function of interest provides more detail about the kinetics of pathogenesis: the rate of release of bacteria, per infected macrophage, as a function of time. The probability that a macrophage, containing $\mathbf{X}_t$ bacteria at time $t$, ruptures before $t + \Delta t$ is $\delta \mathbf{X}_t \Delta t$, in which case the number of bacteria released is simply $\mathbf{X}_t$. The mean number of bacteria released between $t$ and $t + \Delta t$ is therefore $r(t)\Delta t$ where

$$r(t) = \delta \mathbb{E}(\mathbf{X}_t^2 \mid \mathbf{X}_0 = 1) \ . \tag{8}$$

We note that

$$\frac{\mathrm{d}}{\mathrm{d}t} \mathbb{E}(\mathbf{X}_t) = (\beta - \mu)\mathbb{E}(\mathbf{X}_t) - \delta \mathbb{E}(\mathbf{X}_t^2) \ .$$

The most elegant way to evaluate moments of $\mathbf{X}_t$ is making use of the probability generating function:

$$G^{(k)}(z, t) = \sum_{n=0}^{+\infty} p_n^{(k)}(t) z^n \ , \qquad \text{where} \quad p_n^{(k)}(t) = \mathcal{P}[\mathbf{X}_t = n | \mathbf{X}_0 = k] \ . \tag{9}$$

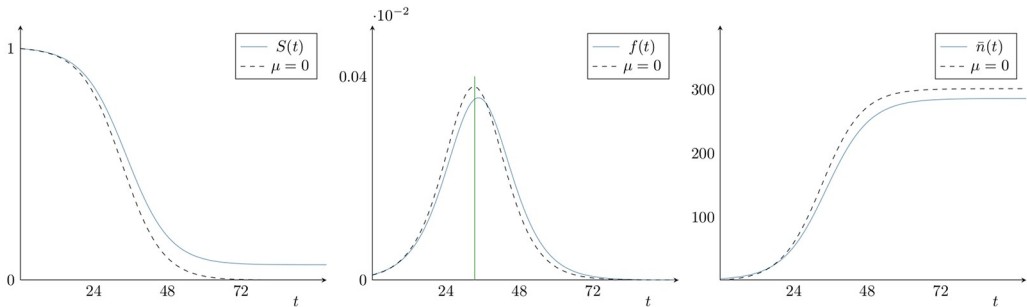

**Fig 5. Survival function, probability density function and bacterial release on rupture.** Left: the macrophage survival probability function, $S(t)$. Centre: the bacterial load is proportional to $f(t) = -\frac{d}{dt}S(t)$. The vertical line is (6). Right: the function $\bar{n}(t)$ that gives the mean number of bacteria released per macrophage. The parameter values, taken from Ref. [23], are $\beta = 0.15\ h^{-1}$, $\mu = 0.01\ h^{-1}$ and $\delta = 0.001\ h^{-1}$. The dashed lines show the corresponding functions when $\mu = 0$.

In Materials and methods, we show that

$$G^{(k)}(z,t) = \left[\frac{ab(1 - e^{-\beta(b-a)t}) + z(be^{-\beta(b-a)t} - a)}{b - ae^{-\beta(b-a)t} - z(1 - e^{-\beta(b-a)t})}\right]^k = \left[\frac{\frac{\mu}{\beta}\gamma(t) + v(t)z}{1 - \gamma(t)z}\right]^k , \qquad (10)$$

where

$$\gamma(t) = \frac{1 - e^{-\beta(b-a)t}}{b - ae^{-\beta(b-a)t}} \qquad \text{and} \qquad v(t) = \frac{be^{-\beta(b-a)t} - a}{b - ae^{-\beta(b-a)t}} .$$

Note that $G^{(k)}(1, t) = S^{(k)}(t)$ [34]. When $\mu = 0$, we have

$$G^{(k)}(z,t) = \left(\frac{ze^{-(\beta+\delta)t}}{1 - zg(t)}\right)^k , \qquad \text{where} \qquad g(t) = \frac{\beta}{\beta + \delta}\left[1 - e^{-(\beta+\delta)t}\right] .$$

In the case $k = 1$, we make use of (10) to obtain $\mathcal{P}[\mathbf{X}_t = 0|\mathbf{X}_0 = 1] = \frac{\mu}{\beta}\gamma(t)$ and the probability that there are $n$ bacteria at time $t$ is

$$\mathcal{P}[\mathbf{X}_t = n|\mathbf{X}_0 = 1] = \left(\frac{b - a}{b - ae^{-\beta(b-a)t}}\right)^2 e^{-\beta(b-a)t}\gamma^{n-1}(t) , \qquad n \geq 1 ,$$

which is a geometric distribution. The function $r(t)$, defined in (8), can be written as the product $f(t)\bar{n}(t)$:

$$r(t) = f(t)\bar{n}(t) = f(t)\frac{1 + \gamma(t)}{1 - \gamma(t)} = \delta\frac{b + 1 - (1 + a)e^{-(\beta+\delta)t}}{b - 1 + (1 - a)e^{-(\beta+\delta)t}} . \qquad (11)$$

Thus, $\bar{n}(t) = (1 + \gamma(t))/(1 - \gamma(t))$, illustrated in Figs 5 and 6, is interpreted as the mean number of bacteria released by a macrophage, given that it ruptures at time $t$. Note that $\lim_{t \to +\infty} \bar{n}(t) = \frac{2\beta + \delta}{\delta}$.

Let $K$ be the random variable denoting the number of bacteria released from a single infected macrophage when it ruptures. If $\mu = 0$ then $\mathbb{E}(K) = (\beta + \delta)/\delta$ [23]. If $\mu \neq 0$ but $\mathbf{X}_t$ is ultimately absorbed into the catastrophe state, the mean number of bacteria released is $b/(b - 1)$ and $\mathcal{P}[K = n]/\mathcal{P}[K = n + 1] = b$ [34]. Karlin and Tavare analysed the processes obtained

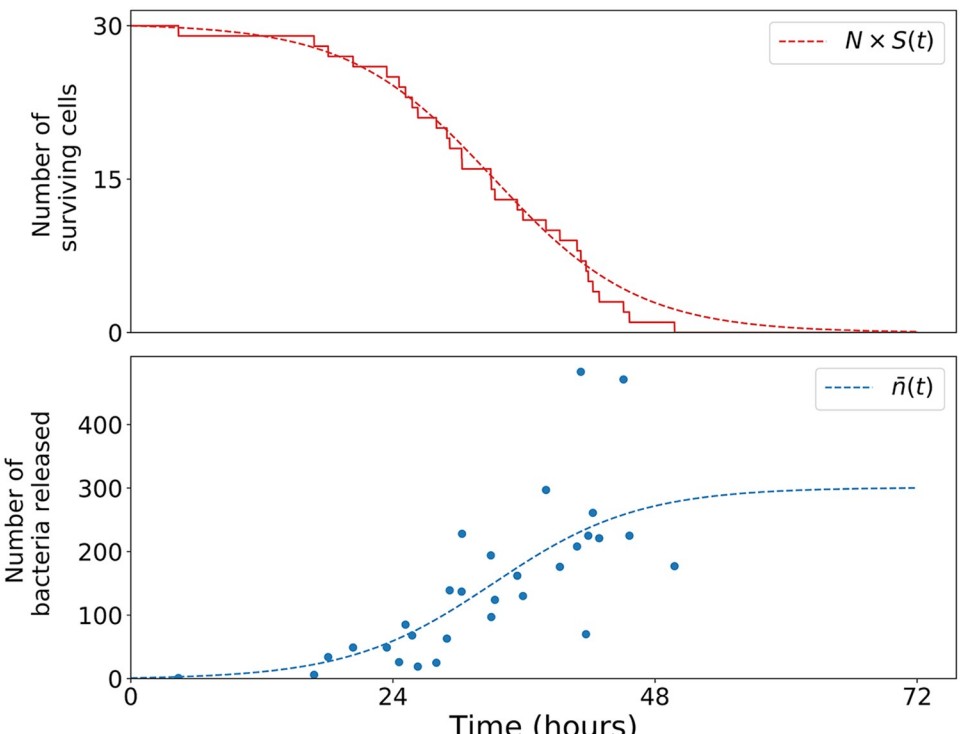

**Fig 6. Agent-based realisation compared to predicted means: First cohort of macrophages.** In a numerical realisation, $N = 30$ macrophages are infected, by one bacterium each, at $t = 0$. The red line shows the number of those macrophages surviving up to time $t$. The dotted red curve is $NS(t)$, using the survival function (5). Each blue dot in the lower panel coincides with a downward step in the red line, corresponding to a macrophage rupture event. The dotted blue curve is $\bar{n}(t)$, given by (11). Parameter values have been taken from Ref. [23]: $\beta = 0.15\ h^{-1}$, $\mu = 0$ and $\delta = 0.001\ h^{-1}$. Agent-based realisations are simulated making use of tau-leaping time-stepping with $\Delta t = 0.01\ h^{-1}$, $M = 10^4$, $\rho = 0.01$ $h^{-1}$, $\phi = 2\ h^{-1}$, $\mu_E = 0.01\ h^{-1}$ and $\gamma = 1\ h^{-1}$.

by conditioning on the ultimate fates: elimination or catastrophe [34]. Taking both possibilities into account, the mean number of bacteria released is

$$\mathbb{E}(K) = \int_0^{+\infty} f(t)\bar{n}(t)\,\mathrm{d}t = \frac{b}{b-1}(1-a) = \frac{\delta b}{\beta(b-1)^2}\ .$$

## Cohort analysis

Bacterial loads can be measured in different organs of a given infected mouse, but only at one time point. In an agent-based simulation, on the other hand, the entire history of every macrophage and bacterium is available. We classify bacteria into cohorts, according to their lineage, assigning a "cohort number" attribute to each bacterium as follows. At the start of the realisation, the cohort number of every bacterium is equal to zero. The cohort number increases by one whenever a bacterium enters a macrophage. When bacteria divide, the daughters inherit their cohort number from their mother. Thus, for the initial dose of bacteria, each bacterium has a cohort number equal to zero. Following their uptake by macrophages, each of their cohort numbers increases to one. A realisation of first cohort macrophage rupture events is shown in Fig 6.

In order to calculate the total number of bacteria in a particular organ, we consider cohorts of bacteria contained within macrophage phagosomes and cytosols. We define the quantities

- $P_n(t)$, the mean number of cohort $n$ bacteria in macrophage phagosomes at time $t \geq 0$, and

- $C_n(t)$, the mean number of cohort $n$ bacteria in macrophage cytosols at time $t \geq 0$.

The initial condition is $P_1(0) = N$. That is, we assume phagocytosis of the initial dose of bacteria occurs instantaneously. Bacteria enter the phagosome from a previous cohort rupture event and escape to the cytosol with rate $\phi$. The bacteria inside the cytosol then replicate with rate $\beta$ before being released in a cohort rupture event. In the calculations of this section, we assume that all bacteria released in macrophage rupture events are immediately absorbed by uninfected macrophages. This assumption is consistent with the dynamics of the agent-based model, as long as the supply of uninfected macrophages is much larger than the number of extracellular (or free) bacteria.

Given the per bacterium rate of phagosomal escape, $\phi$, the mean number of first cohort bacteria in phagosomes is simply

$$P_1(t) = N \mathrm{e}^{-\phi t} \ . \tag{12}$$

In the agent-based computation, the bacteria escape from the phagosome to the cytosol at a different time in each macrophage, drawn from an exponential distribution with mean $1/\phi$. Accounting for this delay, the mean number of first cohort bacteria in macrophage cytosols is, thus, equal to

$$C_1(t) = \int_0^t \frac{\phi}{\delta} P_1(s) f(t-s) \mathrm{d}s = \frac{N\phi}{\delta} \int_0^t f(t-s) \mathrm{e}^{-\phi s} \mathrm{d}s \ .$$

Here, $f(t)/\delta$ is the mean of the birth-death-catastrophe process considered in the previous section, $\mathbb{E}(\mathbf{X}_t)$, where $t = 0$ was taken to be the time of phagosomal escape of the initial bacterium to the cytosol. The mean rate of release of first cohort bacteria from rupturing macrophages at time $t$ after the start of the experiment, is $r_1(t)$ where

$$r_1(t) = N \int_0^t \phi \mathrm{e}^{-\phi s} r(t-s) \mathrm{d}s \ . \tag{13}$$

For the second cohort, the functions $P_2$ and $C_2$ satisfy

$$\frac{\mathrm{d}}{\mathrm{d}t} P_2 = -\phi P_2 + r_1(t) \ , \qquad P_2(0) = 0 \ , \tag{14}$$

and

$$C_2(t) = \frac{\phi}{\delta} \int_0^t P_2(s) f(t-s) \mathrm{d}s \ . \tag{15}$$

If we define

$$r_n(t) = \int_0^t \phi P_n(s) r(t-s) \mathrm{d}s \qquad n = 1, 2, \ldots \ ,$$

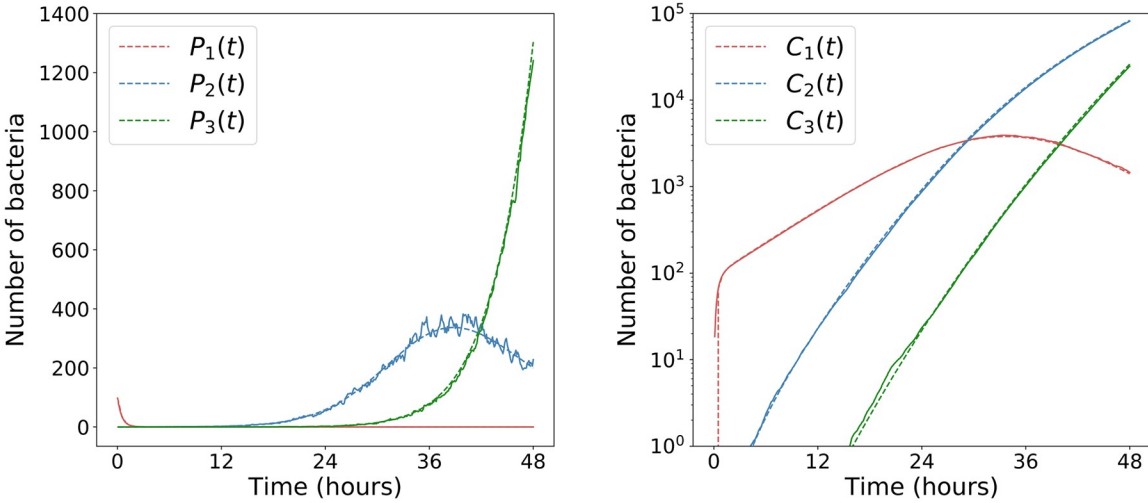

**Fig 7. Cohorts of bacteria in the lung.** The formulæ for the number of bacteria in phagosomes (left) and cytosols (right), obtained from (16a) and (16b) are shown as dashed lines. Averages over $10^2$ realisations of the agent-based computational model are shown as solid lines. The parameter values, taken from Ref. [23], are $\beta = 0.15\ h^{-1}$, $\mu = 0$, $\delta = 0.001\ h^{-1}$, $M = 10^4$, $\rho = 0.01\ h^{-1}$, $\phi = 2\ h^{-1}$, $\gamma = 0.1\ h^{-1}$, $\mu_E = 0.01\ h^{-1}$, $N = 10^2$ and $\Delta t = 0.01h^{-1}$.

then, in general, higher order cohorts of phagosomal and cytosolic bacteria satisfy

$$\frac{\mathrm{d}}{\mathrm{d}t}P_n(t) = -\phi P_n(t) + r_{n-1}(t) , \qquad P_n(0) = 0 \qquad n = 2, 3, \dots , \tag{16a}$$

$$C_n(t) = \int_0^t \frac{\phi}{\delta} P_n(s) f(t-s)\mathrm{d}s , \qquad C_n(0) = 0 \qquad n = 1, 2 \dots . \tag{16b}$$

For the initial 48 hours of *Francisella tularensis* infection, it is sufficient to only consider the first three cohorts of bacteria, or equivalently, first and second order cohort rupture events. A comparison between the mean of $10^2$ simulations of the agent-based model and these approximations is provided in Fig 7. With the cohorts of bacteria in macrophage phagosomes and cytosols, the total number of intracellular bacteria in the lung at time $t$ is found by summing the total number in each cohort.

In order to determine the mean number of bacteria in the MLNs, liver, kidney and spleen, the mean number of extracellular bacteria in the lung must first be calculated. Bacteria released through rupture events either reinfect macrophages in the lung with rate $M\rho\text{(bacteria }h)^{-1}$, are killed extracellularly with rate $\mu_E$, or migrate to a different organ with rate $\gamma$. Here, $\gamma$ is the per bacterium rate of exiting the lung. The destination of a migrating bacterium is then determined by weights assigned to each organ. These weights are given by $w_j$ for $j \in \mathcal{S} = \{liver, MLNs, kidney, spleen\}$, and satisfy $\Sigma_j\, w_j = 1$ [60]. The rate at which a bacterium migrates from the lung to the liver, say, is then $\gamma w_{liver}$. If $E(t)$ denotes the mean number of extracellular bacteria in the lung at time $t \geq 0$, then this variable satisfies

$$\frac{\mathrm{d}}{\mathrm{d}t}E(t) = \sum_{n=1}^{3} r_n(t) - E(t)(M\rho + \gamma + \mu_E) , \quad E(0) = 0 ,$$

where $M$ is the number of macrophages within the lung capable of taking up bacteria.

In the agent-based model, the dynamics in the other organs is equivalent to that in the lung. However, since all bacteria are intracellular until first cohort rupture events occur, and are likely to again infect macrophages in the lung following these events, the mean number of extracellular bacteria is small during the early stages of infection. Assuming that bacteria migrating away from the lung are quickly phagocytosed upon reaching their destination, the mean number of bacteria contained within macrophage phagosomes and cytosols in organs, aside from the lung, then satisfy

$$\frac{\mathrm{d}}{\mathrm{d}t}P^{(j)}(t) = E(t)\gamma w_j - \phi P^{(j)}(t) , \qquad P^{(j)}(0) = 0 , \tag{17a}$$

$$C^{(j)}(t) = \int_0^t \frac{\phi}{\delta} P^{(j)}(s)f(t-s)\mathrm{d}s , \tag{17b}$$

for $j \in \mathcal{S}$. Together, (16a)–(17b) provide an elegant and accurate approach to describe the mean bacterial loads for the first 48 hours in the lungs, as well as in other organs.

## Parameter inference

Total-order Sobol sensitivity indices, presented in Fig 8, quantify the overall effect of a single parameter on model output [61] with respect to total bacterial counts in the lung and MLNs. The parameters were varied over the ranges indicated by the prior distributions in Table 1, with $\phi \in [0.5, 5]$ and $\log_{10} \mu_E \in [-4, -1]$. The parameter $\beta$ is initially the most important, with $\delta$ having an increasing effect at later times. If infected macrophages rupture quickly, bacteria are not able to replicate as effectively in the cytosol and are more frequently found in extracellular environments and in phagosomes, where bacterial replication does not take place. Thus, larger values of $\delta$ are associated with slower bacterial population growth. In addition to $\beta$ and $\delta$, the per bacterium phagocytosis rate, $M\rho$, and the total migration rate, $\gamma$, are important for describing the dynamics outside the lungs. Together with $\mu_E$, they determine with what probability an extracellular bacterium in the lung is killed, migrates to a different organ, or infects a cell in the lung. For the two remaining parameters, $\mu_E$ and $\phi$, the total-order Sobol indices remain low during the initial 48 hours. This allows us to fix their values when performing the

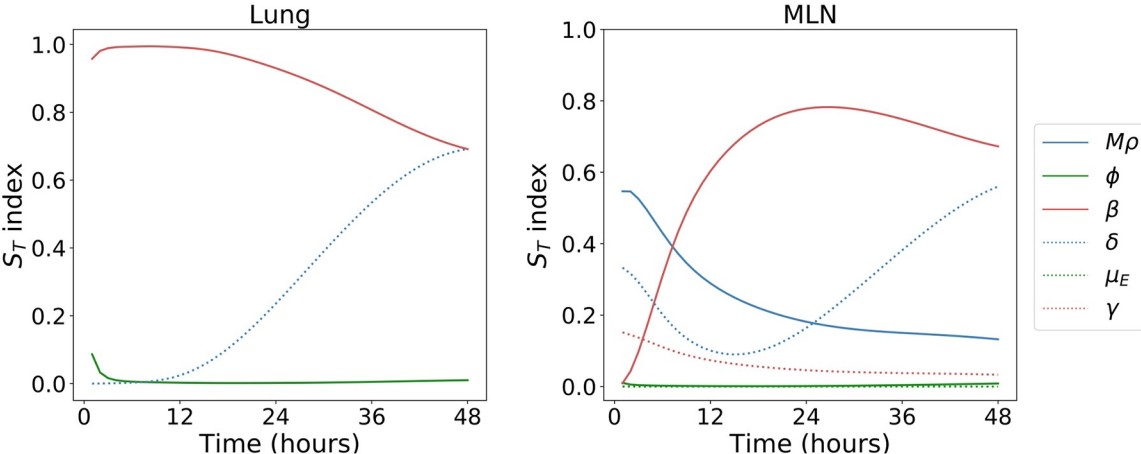

**Fig 8. Total-order Sobol indices.** An indication of the most important parameters that describe the dynamics of total bacterial counts in the lung (left) and MLNs(right) during the first 48 hours of infection. The ranges over which each parameter is varied are: $(M\rho) \in [10^{-2} h^{-1}, 10^5 h^{-1}]$, $\phi \in [0.5h^{-1}, 5h^{-1}]$, $\beta \in [10^{-2} h^{-1}, 1h^{-1}]$, $\delta \in [10^{-5} h^{-1}, 10^{-1} h^{-1}]$, $\mu_E \in [10^{-4} h^{-1}, 10^{-1} h^{-1}]$ and $\gamma \in [1h^{-1}, 10^3 h^{-1}]$.

**Table 1. Agent-based model parameters.** A description and value of the parameters and prior distributions used to determine the total bacterial load in each organ with approximate Bayesian inference. Migration probabilities are calculated using the proportion of bacteria in each organ after 48 hours, starred parameters are inferred using ABC from the observed bacterial counts in each organ.

| Parameter | Description | Prior distribution | Value |
|---|---|---|---|
| $N_{low}$ | low initial dose | - | 2 CFUs |
| $N_{medium}$ | medium initial dose | - | 13.7 CFUs |
| $N_{high}$ | high initial dose | - | 160.33 CFUs |
| $\phi$ | rate of bacterial escape from phagosome | - | $2h^{-1}$ [62] |
| $\mu_E$ | rate of extracellular bacterial death | - | $10^{-2}\ h^{-1}$ [65] |
| $\mu$ | rate of intracellular bacterial death | - | $0h^{-1}$ |
| $w_{\text{MLN}}$ | migration probability to MLNs | - | 0.8 |
| $w_{\text{liver}}$ | migration probability to liver | - | 0.11 |
| $w_{\text{spleen}}$ | migration probability to spleen | - | 0.05 |
| $w_{\text{kidney}}$ | migration probability to kidney | - | 0.04 |
| $M\rho$ | per bacterium infection rate | $\log_{10}(M\rho) \sim U(-2, 5)$ | * |
| $\beta$ | per bacterium replication rate | $\log_{10}\beta \sim U(-2, 0)$ | * |
| $\delta$ | rate of macrophage rupture | $\log_{10}\delta \sim U(-5, -1)$ | * |
| $\gamma$ | per bacterium migration rate | $\log_{10}\gamma \sim U(0, 3)$ | * |

Bayesian inference, with the confidence that any uncertainty in these estimates will have little effect on the total bacterial counts. We therefore only expect to learn about $\beta$, $\delta$, $M\rho$ and $\gamma$, and thus, set $\mu_E = 0.01\ h^{-1}$ and $\phi = 2\ h^{-1}$ [62]. In this section, the decision has also been made to fix the rate of intracellular bacterial death to zero; that is, $\mu = 0$, given the belief that macrophages will likely rupture rather than clear their bacterial load. Including $\mu$ in the inference would also affect our ability to learn about $\beta$, with these two parameters not identifiable from measurements of bacterial counts alone. Finally, the values of the weights that dictate which organs bacteria exiting the lung migrate to are selected, based on the data (summarised in Table 2). displayed in Fig 9. In order to do so we consider the fraction of bacteria that are present in each organ after 48 hours. The mean fraction of bacteria in each organ was used to obtain the following weights: $w_{\text{MLN}} = 0.8$, $w_{\text{liver}} = 0.11$, $w_{\text{spleen}} = 0.05$ and $w_{\text{kidney}} = 0.04$. The larger weight assigned to the MLNsis reasonable, since bacteria may be drained rapidly through the lymphatic system to the MLN [63, 64].

The aim of this section is to make use of the experimental data (see Fig 9) and the mathematical cohort model described in the previous section, to learn about the selected model parameters with an approximate Bayesian computation (ABC) rejection sampling algorithm [42]. Initial uncertainty regarding each parameter is encoded in the prior distributions described in Table 1. For every set of parameters sampled from the prior distributions, the total number of bacteria in each organ is calculated (mod), and compared to the experimental observations (exp) using the distance function

$$d^2(\text{mod}, \text{exp}) = \sum_{i \in \mathcal{D}} \sum_{j \in \mathcal{S}} \sum_{t \in \mathcal{T}_{i,j}} \left[ \frac{\log(B_{i,j}^{(\text{mod})}(t)) - \log(\bar{B}_{i,j}^{(\text{exp})}(t))}{\sigma_{i,j}^{(\text{exp})}(t)} \right]^2,$$

where $\mathcal{D}$ is the set of initial doses and $\mathcal{T}_{i,j}$ is the set of times at which measurements are provided for a given dose $i \in \mathcal{D}$ and organ $j \in \mathcal{S}$. For the lung, model predictions of bacterial burdens, $B_{i,j}^{(\text{mod})}$, are found by summing the initial three cohorts of phagosomal and cytosolic bacteria, (12)–(16b). For each of the remaining organs, where it is assumed that a bacterium is

**Table 2. Table with experimental data sets. Bacterial counts in the lung, MLN, liver, kidney and spleen used for the parameter inference.** Mice are exposed to either 160.33 CFU (top) or 13.7 CFU (bottom) of *Francisella tularensis* SCHU S4 bacteria. Geometric means and standard deviations (SD) are also given.

| time(*hours*) | organ | mouse | | | | | | mean | SD |
|---|---|---|---|---|---|---|---|---|---|
| | | High infectious dose (160.33 CFU) | | | | | | | |
| | | 1 | 2 | 3 | 4 | 5 | 6 | | |
| 1 | lung | 0 | 0 | $2.50 \times 10^2$ | $1.50 \times 10^2$ | 0 | $3.50 \times 10^1$ | $1.05 \times 10^1$ | $1.42 \times 10^1$ |
| 18 | lung | $2.60 \times 10^3$ | $3.25 \times 10^3$ | $1.50 \times 10^3$ | $1.55 \times 10^3$ | $1.20 \times 10^3$ | $2.50 \times 10^3$ | $1.97 \times 10^3$ | 1.48 |
| | MLN | $4.00 \times 10^1$ | $8.50 \times 10^1$ | $5.00 \times 10^0$ | $7.00 \times 10^1$ | $2.45 \times 10^2$ | $6.50 \times 10^1$ | $5.16 \times 10^1$ | 3.64 |
| 24 | lung | $4.95 \times 10^3$ | $3.15 \times 10^3$ | $3.15 \times 10^3$ | $4.90 \times 10^3$ | $1.41 \times 10^3$ | $1.50 \times 10^3$ | $2.83 \times 10^3$ | 1.74 |
| 48 | lung | $2.65 \times 10^5$ | $2.85 \times 10^5$ | $1.30 \times 10^6$ | $1.70 \times 10^5$ | $4.90 \times 10^5$ | $4.25 \times 10^5$ | $3.89 \times 10^5$ | 2.01 |
| | MLN | $2.05 \times 10^3$ | $6.15 \times 10^3$ | $3.40 \times 10^3$ | $1.50 \times 10^2$ | $2.55 \times 10^3$ | $1.20 \times 10^3$ | $1.64 \times 10^3$ | 3.64 |
| | liver | $5.00 \times 10^2$ | $6.00 \times 10^2$ | $4.00 \times 10^2$ | $3.00 \times 10^2$ | $1.00 \times 10^3$ | $6.00 \times 10^2$ | $5.28 \times 10^2$ | 1.51 |
| | kidney | $3.50 \times 10^2$ | $2.00 \times 10^2$ | $3.50 \times 10^2$ | $1.00 \times 10^2$ | $1.00 \times 10^2$ | $1.50 \times 10^2$ | $1.82 \times 10^2$ | 1.77 |
| | spleen | $3.00 \times 10^1$ | $5.00 \times 10^1$ | $8.50 \times 10^2$ | $3.00 \times 10^1$ | $5.50 \times 10^2$ | $5.50 \times 10^2$ | $1.50 \times 10^2$ | 4.94 |
| | | Medium infectious dose (13.7 CFU) | | | | | | | |
| time(*hours*) | organ | 1 | 2 | 3 | 4 | 5 | 6 | mean | SD |
| 1 | lung | $1.00 \times 10^1$ | 0 | 0 | $5.00 \times 10^1$ | 0 | 0 | 2.82 | 5.38 |
| 12 | lung | $5.00 \times 10^1$ | $5.00 \times 10^0$ | 0 | 0 | 0 | $2.00 \times 10^1$ | 4.14 | 5.58 |
| 18 | lung | $2.5 \times 10^2$ | $1.90 \times 10^2$ | $4.35 \times 10^2$ | $1.65 \times 10^2$ | $3.90 \times 10^2$ | $1.50 \times 10^2$ | $2.42 \times 10^2$ | 1.57 |
| 24 | lung | $1.25 \times 10^3$ | 0 | 0 | $3.00 \times 10^2$ | $1.50 \times 10^2$ | $1.85 \times 10^2$ | $4.67 \times 10^1$ | $2.15 \times 10^1$ |
| 48 | lung | $1.10 \times 10^4$ | $2.75 \times 10^4$ | $7.65 \times 10^4$ | $8.85 \times 10^4$ | $2.20 \times 10^4$ | $3.25 \times 10^4$ | $3.37 \times 10^4$ | 2.19 |
| | MLN | $1.95 \times 10^3$ | $4.95 \times 10^2$ | $2.00 \times 10^3$ | $5.35 \times 10^3$ | $2.25 \times 10^2$ | $1.20 \times 10^3$ | $1.19 \times 10^3$ | 3.08 |

only able to infect one macrophage during the initial 48 hours, $B_{i,j}^{(\mathrm{mod})}$ is found by summing (17a) and (17b). For the geometric mean, $\bar{B}_{i,j}^{(\mathrm{exp})}$, and geometric standard deviation, $\sigma_{i,j}^{(\mathrm{exp})}$, experimental observations yielding no bacteria are set to one bacterium.

In total, $10^6$ iterations of the rejection sampling algorithm were performed. The acceptance rate is 0.5%, which leads to an accepted posterior sample of $5 \times 10^3$ parameter sets. Pointwise median predictions, provided in Fig 10, confirm that the posterior samples can reproduce the behaviour experimentally observed during the first 48 hours of infection.

The posterior distribution of $\beta$, shown in Fig 11, is narrow. This indicates that the experimental data together with the mathematical model allows us to learn about this parameter. We find $\beta = 0.154 \pm 0.014 \; h^{-1}$, a range that includes the value considered in Ref. [23] based on reports that the doubling time of a single *Francisella tularensis* bacterium is approximately five hours. The wide posterior distribution of $\delta$, in Fig 11, suggests that it is only possible to identify an upper bound of $\delta = 10^{-2} \; h^{-1}$. However, if $\tau^{\mathrm{rupture}}$ is a random variable for the time a Yule-catastrophe process [34] takes to reach the rupture state $H$, the mean time until rupture is

$$\mathbb{E}(\tau^{\mathrm{rupture}}|\mathbf{X}_0 = 1) = \int_0^{+\infty} f(t)t \, \mathrm{d}t = \frac{1}{\beta}\log\left(\frac{\beta+\delta}{\delta}\right) ,$$

where $f(t)$ is the density of the time until rupture, given in (7). As this is a function of $\beta$ and $\delta$ only, and $\beta$ is confined to a small range of values (as described above), this independent estimate of the mean rupture time allows us to improve our learning about $\delta$.

Wood *et al.*, by comparing to data from an *in vitro* study involving the infection of macrophages with *Francisella tularensis* bacteria, estimate a mean rupture of time of 44.4 *h* [30]. By

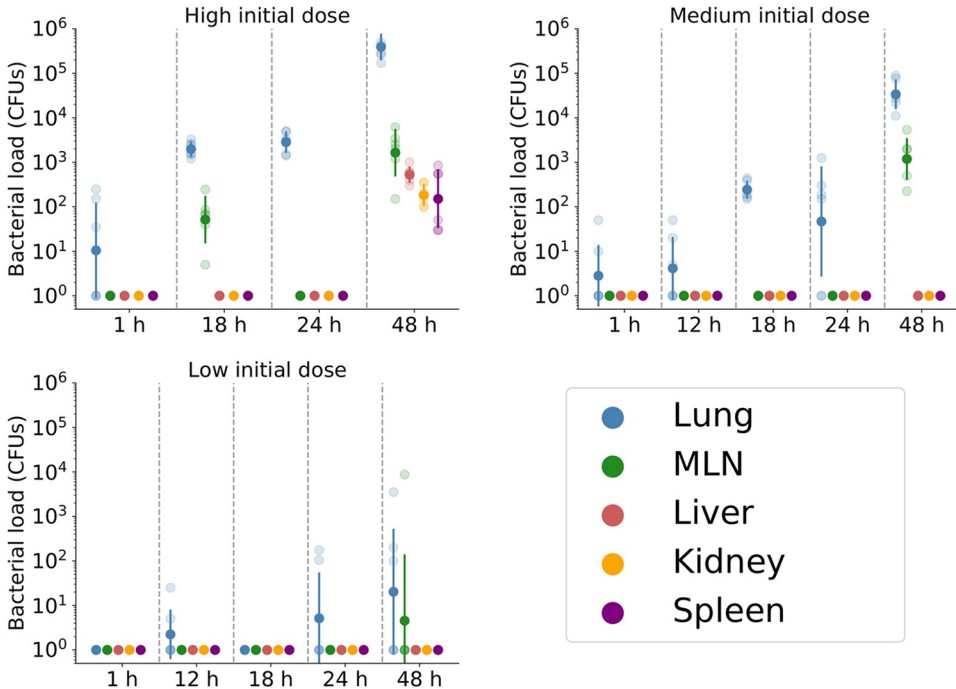

**Fig 9. Bacterial counts in the lung, MLN, liver, kidney and spleen.** Mice are exposed to either 160.33 CFUs (high), 13.7 CFUs (medium) or 2 CFUs (low) of *Francisella tularensis* SCHU S4 bacteria. The observed data are denoted by shaded points, whilst the geometric mean and standard deviation are represented by solid points and bars, respectively. Zero counts have been replaced by one in order to calculate the geometric mean and standard deviation.

choosing pairs $(\beta, \delta)$ such that the mean rupture time is 44.4±0.5 $h$, this additional knowledge allows us to refine the posterior distribution of $\delta$ (see Fig 11 (right)), now yielding a posterior median estimate of $\delta_{\text{median}} \approx 1.5 \times 10^{-4} \, h^{-1}$ and a significantly narrower range of $6.3 \times 10^{-5}$ $h^{-1} \leq \delta \leq 3.8 \times 10^{-4} \, h^{-1}$.

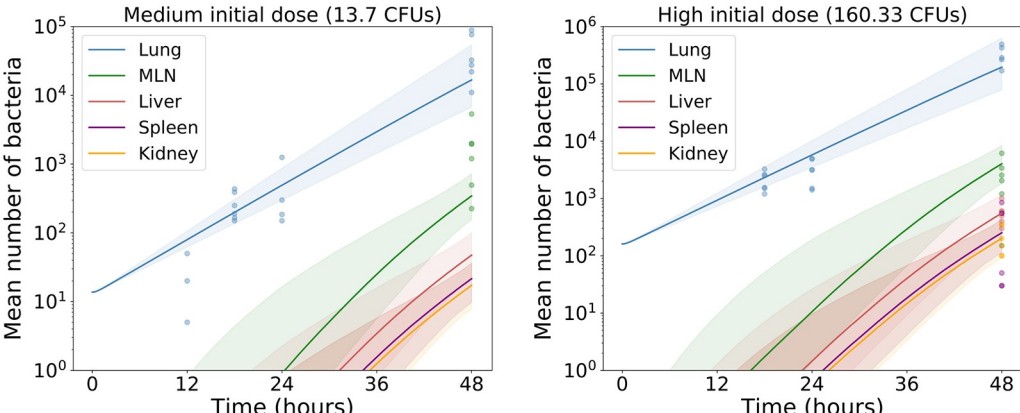

**Fig 10. Pointwise median predictions.** A comparison between model predictions of total bacterial counts and observed bacteria counts for medium (left) and high (right) initial doses. Solid curves and shaded regions, respectively, denote pointwise median predictions and 95% credible regions. These have been constructed using all parameter sets from the posterior sample.

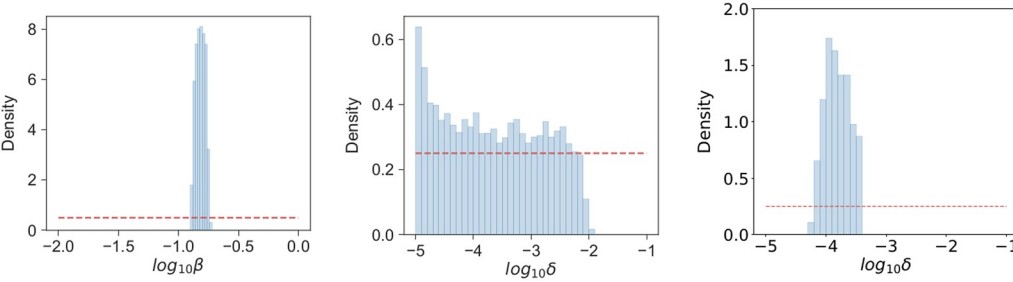

**Fig 11. Posterior histograms for β and δ.** From the posterior sample, the histogram for β (left) shows the significant learning that has been achieved making use of the experimental data. When comparing to bacterial counts, only an upper bound for δ can be identified (centre). However, this distribution can be refined when also considering model predictions for the mean time until rupture (right).

For the parameters $M\rho$ and $\gamma$, refined individual learning is not possible. However, the bivariate posterior histogram in Fig 12 shows that a strong correlation exists between these parameters: increasing $M\rho$ increases the likelihood that an extracellular bacterium again infects a macrophage in the lung, which can be balanced by also increasing the rate at which bacteria leave the lung. Summary statistics for each of the posterior samples are reported in Table 3. A useful quantity is $\frac{\gamma}{(\gamma + M\rho + \mu_E)}$, which is the probability that a bacterium migrates to a different organ, rather than dying or infecting another macrophage in the lung. The corresponding posterior distribution, constructed using the posterior samples of $M\rho$ and $\gamma$, along with $\mu_E = 0.01\ h^{-1}$, is provided in Fig 12. Here, a posterior median value suggests that approximately 4% of extracellular bacteria in the lung are directly involved in the early dissemination (first 48 hours) of *Francisella tularensis* infection to other organs.

It is often difficult to predict the course of infection when infecting mice at low initial doses of bacteria. Here, of the 24 mice infected with 2 CFUs of *Francisella tularensis* bacteria and culled between 12 and 48 hours post-infection, only seven had detectable levels of bacteria present in their lungs. Only one mouse had detectable levels in any of the remaining organs measured. These bacterial counts are presented in Fig 13, alongside model predictions

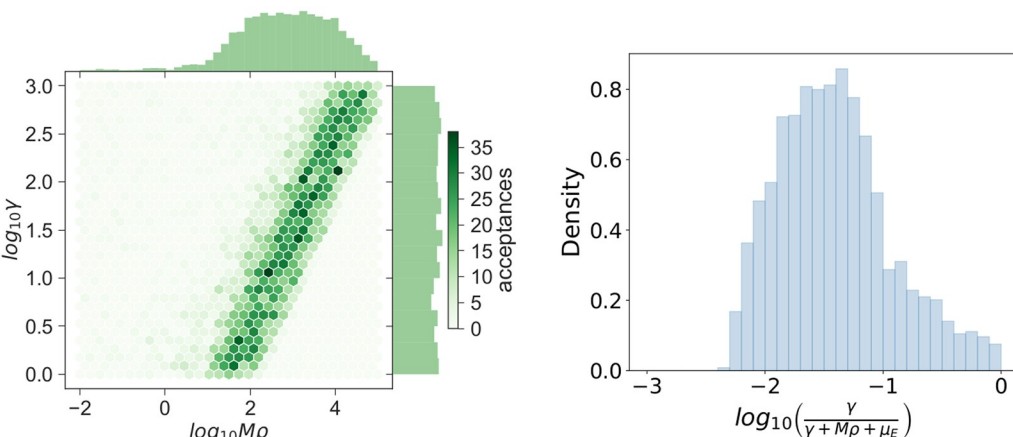

**Fig 12. Posterior histogram for Mρ and γ.** Bivariate histogram (left) depicting the strong correlation between $M\rho$ and $\gamma$ and a histogram of the migration probability (right), constructed using posterior samples of $M\rho$, $\gamma$ and $\mu_E = 0.01\ h^{-1}$.

**Table 3. Summary statistics of the posterior sample for each parameter included in the approximate Bayesian inference.** Posterior samples contain $5 \times 10^3$ values.

| Parameter | Minimum | 1st Quartile | Median | Geo. Mean | 3rd Quartile | Maximum |
|---|---|---|---|---|---|---|
| $\beta$ | $1.29 \times 10^{-1}$ | $1.42 \times 10^{-1}$ | $1.53 \times 10^{-1}$ | $1.53 \times 10^{-1}$ | $1.64 \times 10^{-1}$ | $1.88 \times 10^{-1}$ |
| $\delta$ | $6.31 \times 10^{-5}$ | $1.06 \times 10^{-4}$ | $1.50 \times 10^{-4}$ | $1.55 \times 10^{-4}$ | $2.20 \times 10^{-4}$ | $3.77 \times 10^{-4}$ |
| $M\rho$ | $1.01 \times 10^{-2}$ | $1.19 \times 10^2$ | $7.47 \times 10^2$ | $6.31 \times 10^2$ | $4.64 \times 10^3$ | $9.90 \times 10^4$ |
| $\gamma$ | $1.00 \times 10^0$ | $5.99 \times 10^0$ | $3.27 \times 10^1$ | $3.27 \times 10^1$ | $1.86 \times 10^2$ | $9.99 \times 10^2$ |

obtained using the posterior distributions that were previously inferred from the medium and high infectious dose data. The predictions for the lung agree well with the observed bacterial counts, whilst the predictions for other organs are informative for understanding expected disease progression. More importantly at low initial doses, the stochastic nature of the agent based model described here would allow us to estimate the probability that all bacteria are cleared and the mouse recovers.

## Discussion

There is a tradition of mathematical models that consider populations of infectious agents, target cells and infected cells [24, 25, 66–70]. The usual assumption that new infectious agents are produced at a rate proportional to the number of infected cells, perhaps after an "eclipse" phase [27, 28], may be appropriate in situations where infected cells, independently, release new infectious agents, one or a few at a time, on multiple occasions during their lifetime, a process known as "budding" [32]. It is more problematic when infectious particles are released in a single "burst" as the infected cell dies. The burst scenario is found in many types of infection, including the pathogen of interest in this work, *Francisella tularensis*.

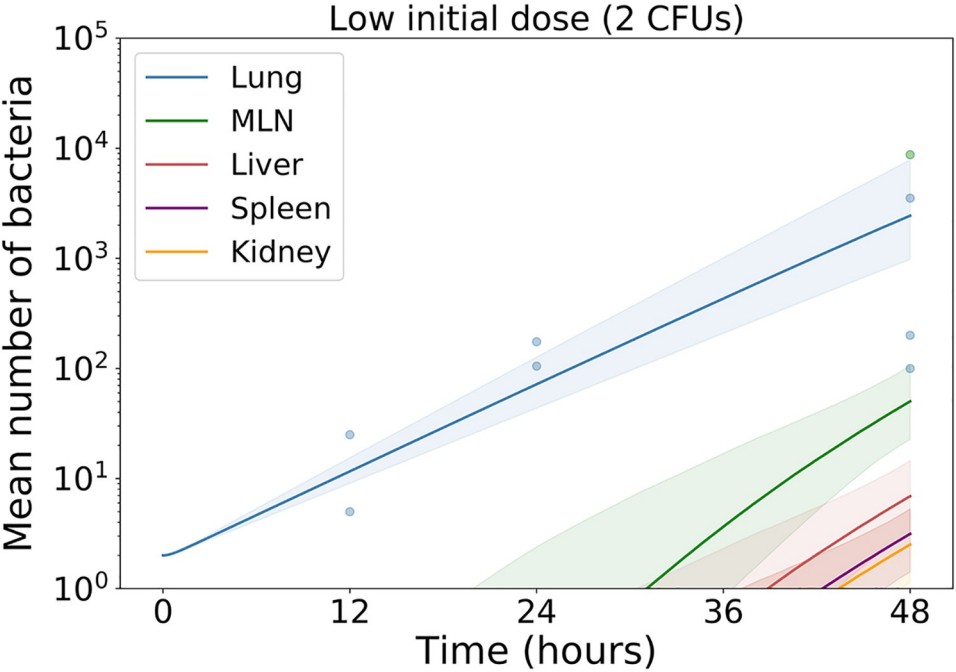

**Fig 13. Low infectious dose predictions.** Predictions of total bacterial counts in each organ following infection at a low initial dose. Posterior distributions inferred by performing ABC with the medium and high doses have been used to create pointwise median predictions and 95% credible regions.

Our approach to understanding the first 48 hours of infection is to focus on dynamics within a single cell. Our within-macrophage model describes the intracellular replication of bacteria and the rupturing and death of the macrophage. Because a rupture event immediately releases all the bacteria from a cell, the stochastic process is a birth-death-catastrophe process (or Yule-catastrophe process if the death rate is zero) [34]. We assume that the rupture rate of a macrophage containing $\mathbf{X}_t$ cytosolic bacteria at time $t$ is $\delta\mathbf{X}_t$. In the first 48 hours post-infection, the bacteria released in catastrophe events rapidly enter new macrophages. Therefore, the growth rate of the total bacterial load is not strongly dependent on this assumption or on the value of $\delta$. We show that $\mathbf{X}_t$ has a geometric distribution, the probability density of macrophage rupture times is $\delta\mathbb{E}(\mathbf{X}_t)$ and the mean rate of release of bacteria, as a function of time, is $\delta\mathbb{E}(\mathbf{X}_t^2)$.

Our modelling approach is applicable to other intracellular pathogens, such as *Salmonella enterica*, and *Bacillus anthracis*, where models must also consider germination of spores [71]. Levofloxacin and ciprofloxacin are antibiotics commonly used to treat tularemia, although their success relies on early administration, which is often made difficult given the non-specific symptoms [72]. Pharmacokinetic and pharmacodynamic models can be used to describe the concentration of antibiotic in each organ and the effect it has on the bacterial load [73]. In epidemic models, birth and death events describe the infection and recovery of animals within a disease reservoir; an analogue of a catastrophe event is the "spillover" of disease into a human population [74]. In models of receptor-mediated signalling events [75], a sequence of states is used to represent reversible phosphorylation events initiated by the binding of a ligand, with dissociation of the ligand leading to the termination of the signal.

In addition to the mathematical analysis focusing on a single cell, we have also described *Francisella tularensis* pathogenesis with an agent-based computational model. Because the history and family tree of every bacterium is available in an agent-based model, it is useful to classify bacteria by cohorts, according to how many different macrophages the bacterium and its ancestors have entered. Knowing the distributions of rupture time and numbers of bacteria released allows us to provide a cohort description of the total bacterial load, in phagosomes and cytosols of infected macrophages, in the lung and in other organs.

With experimental data (see Fig 9) and approximate Bayesian computation, we learn about the parameters of the cohort model. The rate of growth of the experimentally measured bacterial load is primarily determined by the rate of intracellular bacterial division, a parameter that does not appear in a model that only counts numbers of free bacteria and infected cells. Our experimental data, from *in vivo* measurements in lungs, lymph nodes, spleen, kidney and liver, have allowed us to tightly confine the posterior distribution of $\beta$ to a range that is consistent with published estimates based on *in vitro* data. On the other hand, the experimental data do not allow us to determine the mode of migration from the lung to other organs, nor to place tight constraints on the associated timescales.

Data from human or primate infection is even more rare than murine data [76, 77]. In silico models serve as a bridge between animal and human research, with the advantage that human pharmacokinetic and pharmacodynamic parameters can be directly applied. Mathematical models, suitably developed and validated, can provide a suite of tools to estimate the result of experiments, inform their design and extrapolate to humans.

## Materials and methods

### Experimental procedures

Six-to-eight week old female BALB/c mice were challenged (inhalational exposure) with *Francisella tularensis* SCHU S4. In these experiments, mice were infected with either 160 (high), 14

(medium) or 2 (low) colony forming units. At each challenge dose, six mice were culled at 1, 18, 24, 48, 72 and 96 hours and the bacterial burden was measured in the lung, liver, spleen, kidney, MLNsand blood. An additional measurement was taken at 12 hours from mice receiving the low or medium dose. All manipulations were carried out under Advisory Committee for Dangerous Pathogens Level 3 containment conditions in a (Level 3 containment) safety cabinet complying with BS 5726. *Francisella tularensis* SCHU S4 was cultured from frozen stock for two days on blood cysteine glucose agar (BCGA) with cysteine at 37˚C. Subsequently, bacteria were harvested to inoculate 50 ml of modified cysteine partial hydrolysate broth with cysteine and glucose and incubated overnight at 37˚C on a rotary shaker (150 rpm). The suspension was then adjusted using phosphate-buffered saline (PBS) until the optical density at 590 nm was 0.10, where the estimated bacterial density would be $5 \times 10^8$ CFU per ml. Bacterial numbers for challenge were determined on agar following serial dilution (1:10) of samples. The work was conducted under the terms of a licence granted in accordance with the UK Animal (Scientific Procedures) Act, 1986. Female BALB/c mice (Charles River Laboratories Ltd, Margate, Kent, UK) were habituated to the experimental animal unit for one week prior to challenge. Environmental conditions were maintained at 21˚C ± 2˚C and 55% ± 10% humidity with lighting set to mimic a 12/12 (hour) dawn to dusk cycle. The mice were then transferred to a Level 3 containment rigid isolator for a further 5 and 7 days. They were housed in polycarbonate cages (six animals per cage) with steel cage tops and corncob bedding (International Product Supplies, Wellingborough, UK). The mice were fed a Teklad TRM 19% protein irradiated diet ad libitum (Harlan Teklad, Bicester, UK) and given fresh water daily.

Mice were challenged by aerosol, using a Henderson-type apparatus [78] and a Collison nebuliser [79]. Briefly, 10 ml of *Francisella tularensis* SCHU S4 culture was aerosolised using a Henderson Apparatus over an exposure time of 10 minutes [80]. The aerosol was delivered at a flow rate of 12 L/min with impinger samples from the exposure apparatus plated on BCGA to calculate retained dose. Using the known flow rate of the Henderson exposure apparatus (66 L/min), bacterial counts from these samples were then converted to bacterial counts per litre of air. The breathing rate of the animals in the apparatus (approximately 20 ml of air per minute) was then added to the calculation along with the length of exposure (10 minutes) to yield an estimated delivered dose expressed in CFU per animal. Previous studies have determined that aerosol uptake in obligate nasal breathers, such as the mouse, is approximately 40% [81]. Using this conversion factor, the estimated retained dose was calculated for each exposure. Mice were culled for analysis of tissues at different time points and exsanguinated using cardiac puncture following terminal anaesthesia. Blood was placed into 1.5 ml heparin tubes. Lungs, spleen and liver were removed and placed into bijoux tubes filled with 2 ml of PBS. Duplicate experiments were performed. All procedures and housing were in accordance with the Animal (Scientific Procedures) Act (1986). Organs were processed at less than 1 h postmortem. Blood was diluted 1:10 in PBS. Collected organs were placed into 6-well trays containing 40 $\mu$m cell sieves with 1,800 $\mu$l of PBS, then disrupted through the cell sieve using the plunger of a 2 ml syringe. Cell suspensions were collected. 100 $\mu$l aliquots of the cell suspension or blood were used for enumeration of bacteria on agar plates following serial dilution in PBS.

## Agent-based model

In our computational model, each macrophage and each free bacterium has a unique identity and set of mutable attributes. The attributes of a macrophage are: spatial location, state of activation, cohort counter (see Cohort analysis Section), and list (and number) of phagosomal and cytosolic bacteria. The spatial location is either lung, liver, spleen, MLNsor kidney. Although bacterial counts have also been measured in the blood, the numbers were small enough (<10

CFUs) that this compartment can been neglected in the model, as a first approximation. Once phagocytosed, a bacterium remains in a macrophage's phagosome for an exponentially-distributed time with mean $1/\phi$, then escapes to the macrophage's cytosol. There, it becomes the founder of a population of intracellular bacteria, governed by the birth-death-catastrophe process, that lasts until either the bacterial population is eliminated from the macrophage, or the macrophage ruptures and releases its contents. Newly-released bacteria suffer one of three fates: phagocytosis by a macrophage in the same organ, death or migration to a different organ. Given that the initial number of resting macrophages is much larger than the initial number of bacteria, events in which an infected macrophage is reinfected by another bacterium are rare in the first 72 hours of pathogenesis.

Macrophages may exhibit a variety of activation states in different tissues [71, 82–85]. At any time in our computational model, each macrophage in a computational realisation is in one of three states: resting, suppressed (anti-inflammatory) or activated (pro-inflammatory). Every one of the initial $M$ macrophages begins in the resting state. On phagocytosis, resting macrophages enter a suppressed state in which they are unable to kill bacteria and secrete the anti-inflammatory cytokine TGF-$\beta$ that contributes to the suppression of other macrophages. Resting macrophages can become activated through the detection of host damage molecules released by rupturing macrophages [22]: each rupture event can result in the activation of a resting macrophage in the same organ. Activated macrophages kill the bacteria they phagocytose. They also secrete IFN-$\gamma$ that provokes activation of neighbouring macrophages.

A numerical realisation starts with the arrival of a number, chosen from a Poisson distribution with mean $N$, of free bacteria in the lung (see Table 1 for values of $N$). Individual rupture times are recorded, and we explicitly track cohorts of bacteria by assigning a "cohort number" attribute to each bacterium (see Cohort analysis Section). Similarly, the set of bacteria inside each macrophage is subdivided by cohort number and each rupture event is classified according to the minimum cohort number of the bacteria released. Computer codes (in *Python*) to generate the numerical realisations of the agent-based model and to perform the cohort analysis are available in this link http://review.researchdata.leeds.ac.uk/id/eprint/1399/.

There are nine types of events in the computational agent-based model: phagocytosis, escape, division, intracellular or extracellular death of a bacterium, rupture, migration, cytokine-mediated suppression or activation of a macrophage. The rates are as follows:

- $\rho$ is the rate of phagocytosis per macrophage,

- $\phi$ is the rate at which bacteria escape the phagosome,

- $\beta$ and $\mu$ are the birth and death rates of bacteria in the cytosol,

- $\mu_E$ is the death rate of free (or extracellular) bacteria,

- $\delta \mathbf{X}_t$ is the rupture rate of a macrophage containing $\mathbf{X}_t$ cytosolic bacteria at time $t$, and

- $\gamma$ is the rate at which bacteria migrate to other organs.

Cytokine-mediated activation and suppression of macrophages are included in the computational model by means of two dimensionless functions of time, $G(t)$ and $T(t)$, in each organ. The first summarises the levels of inflammatory cytokines, such as IFN$\gamma$; the second summarises the levels of anti-inflammatory cytokines, such as TGF-$\beta$. The functions are updated according to the following differential equations

$$\frac{\mathrm{d}}{\mathrm{d}t} G(t) = \alpha_G \, M_A(t) - \mu_G \, G(t) \qquad \text{and} \qquad \frac{\mathrm{d}}{\mathrm{d}t} T(t) = \alpha_T \, M_S(t) - \mu_T \, T(t),$$

where $M_A(t)$ and $M_S(t)$ are the numbers of activated and suppressed macrophages at time $t$, respectively. The parameters have been chosen as follows: $\alpha_G = 10^{-3}\ h^{-1}$ and $\alpha_T = 10^{-1}\ h^{-1}$ are the per macrophage production rates of TGF-$\beta$ and IFN$\gamma$, respectively; the decay rates are $\mu_G = 9 \times 10^{-2}\ h^{-1}$ and $\mu_T = 10^{-1}\ h^{-1}$, respectively [86]. At any time when $T(t)$ exceeds the threshold level $10^2$ in any organ, each resting macrophage in that organ has a rate $\nu_\gamma = 0.04 h^{-1}$ of transition to the suppressed state. At any time when $G(t)$ exceeds the threshold level $10^2$ in any organ, each resting macrophage in that organ has a rate $\nu_\beta = 0.01 h^{-1}$ of transition to the activated state. Although cytokine-mediated events have been included in the computational agent-based model, their effect in *Francisella tularensis* infected mice is minimal during the initial 72 hours [87], and thus, they are not included in the subsequent approximations or parameter inference. Despite this, future experimental measurements of the concentration of these cytokines would enable us to use this agent-based model, along with any learning about the remaining model parameters achieved here, to obtain more accurate estimates of parameters, such as the level of IFN-$\gamma$ required for macrophage activation.

Two types of time-stepping are available:

- The Gillespie algorithm, where the time increments are inter-event times which are drawn from exponential distributions; the probability of each type of event is proportional to its rate, and all rates are updated after each event [23].

- Tau-leaping, where the time increment (or step size), $\Delta t$, is fixed; the number of occurrences of each type of event per step is a Poisson random variable with mean proportional to its rate [88].

When results from agent-based simulations are reported here, the tau-leaping procedure has been applied with a step size (or time increment) of $\Delta t = 10^{-2}$ hours. Due to the large number of macrophages present at the start of the simulation, along with the rapid growth of the bacterial population, the number of agents is too large for the Gillespie algorithm to be implemented efficiently.

## Solution of the probability generating function of a birth-and-death process with catastrophe

Let $G(z, t)$ denote the probability generating function of a birth-and-death process with catastrophe,

$$G(z, t) = \sum_{n=0}^{+\infty} z^k \Pr(\mathbf{X}_t = n \,|\, \mathbf{X}_0 = 1).$$

We then have

$$\frac{\partial G}{\partial t} = [\beta z^2 - (\beta + \mu + \delta)z + \mu]\frac{\partial G}{\partial z}, \qquad G(z, 0) = z.$$

Using the method of characteristics, we may write

$$G(z, t) = f\left(\frac{z - a}{z - b}\mathrm{e}^{-\beta(b-a)t}\right).$$

Given the initial condition $G(z, 0) = z$ and the substitution $\xi = (z - a)/(z - b)$, we find

$$f(\xi) = \frac{\xi b - a}{\xi - 1}\ .$$

Setting $\xi = \mathrm{e}^{-\beta(b-a)t}(z-a)/(z-b)$, one obtains

$$
\begin{aligned}
G(z,t) \quad &= \frac{b(z-a)\mathrm{e}^{-\beta(b-a)t} - a(z-b)}{(z-a)\mathrm{e}^{-\beta(b-a)t} - (z-b)} \\
&= \frac{ab(1 - \mathrm{e}^{-\beta(b-a)t}) + z(b\mathrm{e}^{-\beta(b-a)t} - a)}{b - a\mathrm{e}^{-\beta(b-a)t} - z(1 - \mathrm{e}^{-\beta(b-a)t})} \; .
\end{aligned}
$$

If the process instead starts with $\mathbf{X}_t = k$, then $G^{(k)}(z, 0) = z^k$ and the solution is $G^{(k)}(z, t) = [G(z, t)]^k$.

## Acknowledgments

We acknowledge and are grateful to a number of technical staff and scientists who assisted in performing these experimental procedures. Numerical work was undertaken on ARC3, which is part of the High Performance Computing facilities at the University of Leeds, UK. We acknowledge and are grateful to the International Centre for Mathematical Sciences (ICMS), Edinburgh, where this manuscript was completed during a Research in Groups programme (JC, GL, MLG, TRL and CMP). JC, MLG and CMP acknowledge the support and hospitality of ICTS, Bangalore, India, where the final version of this manuscript was discussed and completed. Content includes material subject to Dstl Crown copyright (2019).

## Author Contributions

**Conceptualization:** Jonathan Carruthers, Grant Lythe, Martín López-García, Joseph Gillard, Thomas R. Laws, Carmen Molina-París.

**Formal analysis:** Jonathan Carruthers, Grant Lythe.

**Investigation:** Jonathan Carruthers, Grant Lythe, Martín López-García, Joseph Gillard, Thomas R. Laws, Roman Lukaszewski, Carmen Molina-París.

**Methodology:** Jonathan Carruthers, Grant Lythe, Martín López-García, Carmen Molina-París.

**Writing – original draft:** Jonathan Carruthers, Grant Lythe, Martín López-García, Carmen Molina-París.

**Writing – review & editing:** Jonathan Carruthers, Grant Lythe, Martín López-García, Joseph Gillard, Thomas R. Laws, Carmen Molina-París.

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
