## [Decision Letter · Decision Letter 0]

5 Nov 2019

Dear Dr molina-paris,

Thank you very much for submitting your manuscript, 'Stochastic dynamics of Francisella tularensis infection and replication', to PLOS Computational Biology. As with all papers submitted to the journal, yours was fully evaluated by the PLOS Computational Biology editorial team, and in this case, by independent peer reviewers. The reviewers appreciated the attention to an important topic but identified some aspects of the manuscript that should be improved.

We would therefore like to ask you to modify the manuscript according to the review recommendations before we can consider your manuscript for acceptance. Your revisions should address the specific points made by each reviewer and we encourage you to respond to particular issues Please note while forming your response, if your article is accepted, you may have the opportunity to make the peer review history publicly available. The record will include editor decision letters (with reviews) and your responses to reviewer comments. If eligible, we will contact you to opt in or out.raised.

- Supporting Information uploaded as separate files, titled 'Dataset', 'Figure', 'Table', 'Text', 'Protocol', 'Audio', or 'Video'.

We hope to receive your revised manuscript within the next 30 days. If you anticipate any delay in its return, we ask that you let us know the expected resubmission date by email at ploscompbiol@plos.org.

Sincerely,

James O'Dwyer

Associate Editor

PLOS Computational Biology

Jason Papin

Editor-in-Chief

PLOS Computational Biology

[LINK]

Reviewer's Responses to Questions

**Comments to the Authors:**

Reviewer #1: I read and assessed this manuscript as a general disease ecologist with a mathematical perspective. I would first like to congratulate the authors on a nicely written manuscript; the biology of the system and the model were well-described and highly accessible.

The authors model the growth of intracellular and extracellular populations of Francisella tularensis following initial host exposure. The primary improvement on previous models of this system is to model cell rupture as a catastrophic event; by subjecting intracellular bacterial populations to a birth-death-catastrophe process, their model introduces stochasticity and strong variation into the growth of the total bacterial population. The authors compare their mathematical results with results from an agent-based model and experimental time series data. They describe the dynamics of infected cells and the bacterial population and find that β (intracellular birth rates) is a critical parameter regulating the system. The manuscript is a nice demonstration of how within-cell processes affect between-cell transmission and overall infection dynamics. I have a few suggestions that I believe will improve the overall impact of the paper.

General comments:

1. Table 1 would be much better as a figure; it contains key empirical data that support the modeling results. As it is presented, the scientific notation makes it difficult to immediately and easily compare among treatments, organs and time periods. Plotting the average value along with SD would eliminate the need to show values for each of the six individuals. Also, why aren’t the low dose data presented here?

2. Lines 114-117 and 284-286: The authors state that the ratio of macrophages to extracellular bacteria is sufficiently high in the first 72 hours that co-infection of a macrophage does not occur. However, bacterial migration to other tissues does occur during this time period, which requires a high number of extracellular bacteria (escaping macrophages). It seems that, if extracellular bacteria populations are sufficiently high for migration, there should be an increasing probability of co-infection (indeed Fig. 11 shows that infection rates and migration are positively correlated). It would be helpful to understand the densities and interactions of these two players by plotting their populations through time. Seeing the ratio of macrophages to extracellular bacteria should illuminate when migration can occur and whether co-infection is possible.

3. In figures 9 and 10 (at high, medium and low initial doses) the modeling results demonstrate earlier arrival of bacteria into non-lung tissues than the empirical work supports. In lines 370-380 this is attributed to bacterial detection, but it could also be that the model results are off. Can this section describe any limitations to the model that might be leading to higher than observed migration rates?

4. The discussion could also better tie the results into the implications for this disease. The authors open the manuscript by discussing tularemia as an extremely infectious pathogen and potential biothreat agent, but the discussion never interprets the results in this context. Three important conclusions of the work are that β has strong effects on the system, that the time between rounds of bacterial division is 6 hours, and that there is a reasonably high rate of migration from lung to additional tissues. How do these results matter for progression of tularemia in humans? The empirical results are from a mouse-model, is there work on human tularemia that supports the conclusions?

5. I found the discussion of the work to be fairly narrow and focused only on the Francisella system. The birth-death-catastrophe process is a novel and exciting component of this work. The authors mention that this process is important for other intracellular pathogens, but those other systems are never actually described or discussed. Given the insights gained here, how might the birth-death-catastrophe process inform disease dynamics in other systems?

Minor comments:

Lines 34-38: Is this there any empirical evidence supporting the load-dependent hazard rate? (It seems biologically reasonable, but empirical support would strengthen this key assumption)

Line 59-61: Clarify here that extracellular bacteria in the blood were also measured.

In all figure legends, it would be helpful to redefine the parameters and variables as much as possible so that the reader doesn’t have to flip back to earlier text/tables to interpret them. The authors have done this in some, but not all, figure legends.

Reviewer #2: Carruthers et al. constructed stochastic models for the infection and replication dynamics of Francisella tularensis. The authors then compared model simulation with experimental data and quantified key parameters for the life cycle of the bacteria in vitro. Francisella tularensis is an important pathogen because of its virulence and transmissibility. However, our quantitative understanding of the life-cycle of the pathogen is very limited. This may hinder the development of effective drugs or vaccines. Thus, the work addresses an important question. The mathematical analysis is rigorous. The methodology is novel and can potentially applicable for other pathogens. The manuscript is well written. I recommend publication once the following concerns are addressed.

First, the paper emphasizes very much on descriptions of the mathematical methodology and the model. However, the implications of the model and how the model results would be useful for addressing biological questions or clinical questions, for example, developing therapeutics, are not well discussed. I think one or two paragraphs of discussion of how the results of this study relates to the implications of the work would substantially increase the relevance of the work.

Second, for a lot of the parameters in the model (e.g. Table 2), it is not clear how those parameter values were derived; or whether it is consistent with what we know about the life cycle of the pathogen. I understand the precise values may be hard to get due to lack of experimental studies, but it would be important to test the sensitivities of the conclusions of the work against uncertainties in these parameter values.

Third, in the introduction (Page 2), the authors claim that classical models ‘assume that each infected cell, independently, releases infectious particles at a constant rate’. The statement is ok given most of the previous models indeed made those assumptions. However, this statement also ignores a recent work by Koelle, et al. Virus Evolution 2019; 5 (2), vez018, where this assumption can be relaxed. I think that work is directly related to the model developed in this manuscript and it is important to include it in the introduction of the literature.

**Have all data underlying the figures and results presented in the manuscript been provided?**

Reviewer #1: Yes

Reviewer #2: Yes

PLOS authors have the option to publish the peer review history of their article (what does this mean?). If published, this will include your full peer review and any attached files.

Reviewer #1: No

Reviewer #2: No

---

## [Decision Letter · Decision Letter 1]

27 Feb 2020

Dear Professor molina-paris,

We are pleased to inform you that your manuscript 'Stochastic dynamics of Francisella tularensis infection and replication' has been provisionally accepted for publication in PLOS Computational Biology.

Before your manuscript can be formally accepted you will need to complete some formatting changes, which you will receive in a follow up email. A member of our team will be in touch within two working days with a set of requests.

Best regards,

James O'Dwyer

Associate Editor

PLOS Computational Biology

Jason Papin

Editor-in-Chief

PLOS Computational Biology

Reviewer's Responses to Questions

**Comments to the Authors:**

Reviewer #1: This is my second review of “Stochastic dynamics of Franciscella tularensis infection and replications”. The authors have addressed all of my major/minor comments and included a new figure (Figure 2) that I found very helpful for interpretation of the empirical results and alignment with the modeling results. I am happy to recommend publication for the manuscript and I have no additional concerns/comments.

Reviewer #2: All my comments are sufficiently addressed.

**Have all data underlying the figures and results presented in the manuscript been provided?**

Reviewer #1: Yes

Reviewer #2: None

PLOS authors have the option to publish the peer review history of their article (what does this mean?). If published, this will include your full peer review and any attached files.

Reviewer #1: No

Reviewer #2: Yes: Ruian Ke

---

## [Editor Report · Acceptance letter]

22 May 2020

PCOMPBIOL-D-19-01195R1 

Stochastic dynamics of Francisella tularensis infection and replication

Dear Dr molina-paris,

I am pleased to inform you that your manuscript has been formally accepted for publication in PLOS Computational Biology. Your manuscript is now with our production department and you will be notified of the publication date in due course.

With kind regards,

Matt Lyles
